


# The impacts of elevated CO₂ on forest growth, mortality and recovery in the Amazon rainforest

Yitong Yao[1,2], Philippe Ciais[1], Emilie Joetzjer[3], Wei Li[4], Lei Zhu[1,4], Yujie Wang[2], Christian Frankenberg[2,5], Nicolas Viovy[1]

[1]Laboratoire des Sciences du Climat et de l'Environnement, LSCE/IPSL, CEA-CNRS-UVSQ, Université Paris-Saclay, Gif-sur-Yvette, 91191, France
[2]Division of Geological and Planetary Sciences, California Institute of Technology, Pasadena, CA 91125, USA
[3]INRAE, Universite de Lorraine, AgroParisTech, UMR Silva, Nancy, 54280, France
[4]Department of Earth System Science, Ministry of Education Key Laboratory for Earth System Modeling, Institute for Global Change Studies, Tsinghua University, Beijing, 100084, China
[5]Jet Propulsion Laboratory, California Institute of Technology, Pasadena, CA 91109, USA

*Correspondence to*: Yitong Yao (yyao2@caltech.edu)

**Abstract.** The Amazon rainforest plays a crucial role in global carbon storage, but a minor destabilization of these forests could result in considerable carbon loss. Among the external factors affecting vegetation, elevated CO₂ (eCO₂) levels have long been anticipated to have positive impacts on vegetation, including direct photosynthesis / productivity enhancement and increasing water use efficiency. However, the overall impact of eCO₂ on the net carbon balance, especially concerning tree mortality-induced carbon loss and recovery following extreme drought events, has remained elusive. Here, we use a process-based model that couples physiological CO₂ effects with demography and drought mortality / resistance processes. The model was previously calibrated to reproduce observed drought responses of Amazon forest sites. The model results, based on factorial simulations with and without eCO₂, reveal that eCO₂ enhances forest growth and promotes competition between trees, leading to more natural self-thinning of the forest stands, following a growth-mortality trade-off response although the growth outweighs the tree loss. Additionally, eCO₂ provides water-saving benefits, reducing the risk of tree mortality during drought episodes, although extra carbon losses still could occur due to eCO₂ induced increase in background biomass density, thus 'more carbon available to lose' when severe droughts happen. Furthermore, we found that eCO₂ accelerates the drought recovery and enhances drought resistance and resilience. These findings illuminate the intricate ways in which increasing CO₂ concentrations shape forest carbon dynamics, offering valuable insights into the evolution of the Amazon rainforest.

## 1 Introduction

The intact Amazon rainforest influences present and future global carbon dynamics, accounting for a current carbon sink of 0.42-0.65 PgC yr⁻¹ for 1990-2007 (Pan et al., 2011), and containing 40% of the tropical forest aboveground biomass (Liu et al., 2015), a large carbon stock that is projected to be vulnerable to climate change (Boulton et al., 2022). Preserving



this carbon stock is essential for regulating global $CO_2$ levels and stabilizing the Earth's climate. As climate change progresses and $CO_2$ levels rise, tropical rainforests can both increase carbon sequestration and become destabilized by climate risks. The impact of elevated $CO_2$ (e$CO_2$) on carbon sequestration separates into direct effect related to higher leaf carboxylation rates, that may translate into higher leaf area index, tree productivity and biomass, known as $CO_2$ fertilization as well as indirect effects of partial stomatal closure and subsequently increased water use efficiency (WUE, $CO_2$ physiological forcing) (Smith et al., 2020). In turn, higher leaf area increases transpiration and can increase water stress. However, the potential translation of increased individual tree growth rates into biomass accumulation at ecosystem level remains uncertain, given that e$CO_2$ not only enhances carbon inputs at ecosystem level, but also amplifies carbon loss, through growth-mortality trade-offs with higher growth possibly leading to more competition between trees and higher mortality rates. Such 'high gain high loss' patterns reflecting a coupling between growth and mortality have been identified across spatial gradients (Needham et al., 2020; Stephenson & van Mantgem, 2005; Walker et al., 2021) and also seem to emerge in terms of temporal trajectories, with a parallel increase of growth and mortality, observed e.g. in the Amazon (Lewis et al., 2004) and other rainforests from long term inventories (Hubau et al., 2020). For example, repeated census of forest inventory plots within intact tropical forests in Amazonia have found a faster increase in carbon losses from tree mortality, which surpasses the increase in carbon gains attributed to both tree growth and new tree recruitment, resulting in a decline in the net forest carbon sink (Hubau et al., 2020). Although a positive effect of e$CO_2$ on increased tree loss has been hypothesized by McDowell et al (2022), establishing a significant correlation between carbon loss and e$CO_2$ has proven to be elusive (Hubau et al., 2020).

Compared to the research concentrating on vegetation productivity in response to e$CO_2$, less attention has been directed toward the response of carbon loss, although minor disruptions of mortality rates in high-biomass systems like the Amazon intact rainforests could trigger substantial carbon loss. An increase in tree mortality can reduce the plant carbon residence time and consequently counteract the enhanced productivity (Friend et al., 2014). A comprehensive understanding of the response of tree mortality to e$CO_2$ is thus crucial for unravelling the forest biomass carbon dynamics. Carbon loss can arise from internal ecosystem processes like competition-induced self-thinning and death of demoted trees, death of individual large trees forming gaps, as well as from external drivers such as extreme climate events, insects, and pathogens (Fig. 1). McDowell et al (2018) outlined two potential mechanisms underlying the connection between e$CO_2$ and an increased tree mortality rate. First, enhanced individual tree growth rates could accelerate self-thinning due to increased competition. Second, e$CO_2$ makes trees larger and more vulnerable to external environmental conditions of wind damage, drought, and heat (Gora & Esquivel-Muelbert, 2021; Maia et al., 2020). These two mechanisms correspond to competition-induced carbon loss and drought-induced carbon loss, and they pose threats on smaller trees and larger trees, separately. But e$CO_2$ also has the potential to promote tree survival by improving water use efficiency during drought (Brienen et al., 2017a, 2017b; Van der Sleen et al., 2015). Liu et al (2017) demonstrated, through simulations with a detailed soil-plant hydraulic model, that e$CO_2$ mitigates drought risks by decreasing the fraction of days when the daily minimum xylem water potential



is below a critical threshold. Findings from a global model simulating hydraulics and demography conducted by Yao et al (2023) also indicated that drought exposure could be alleviated under eCO$_2$ in the Amazon. Besides, eCO$_2$ effects are also regulated by hydro-climatic conditions. Fatichi et al (2016) revealed that indirect effects on productivity from eCO$_2$ tend to be more pronounced in water-limited ecosystems although severe water stress can offset the expected CO$_2$ fertilization

effects (Kolby Smith et al., 2015). The magnitude of the water saving effect is also modulated by the intensity and duration of water stress events (Birami et al., 2020). Therefore, given the interplay of enhanced photosynthesis, heightened competition, vulnerability due to larger size, and mitigating effect from water saving benefits, the impact of eCO$_2$ on carbon balance is not a straightforward monotonic relationship. The relative rates at which gross carbon fluxes change with eCO$_2$ play a crucial role in determining the net changes in AGB (Fig. 1).


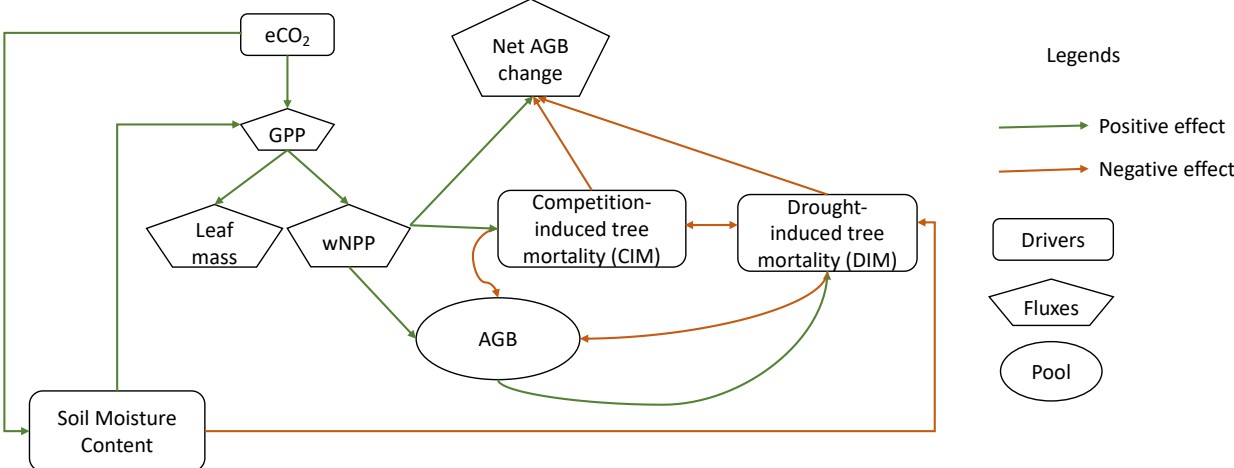

**Figure 1.** A schematic plot illustrating the influence of eCO$_2$ on forest biomass carbon balance. A positive effect means that the impacted variable is expected to increase from eCO$_2$ as compared to pre-industrial CO$_2$. A positive effect on net AGB change means that eCO$_2$ leads to a higher AGB net carbon gain. When CO$_2$ concentration rises, water use efficiency

increases due to partial closure of stomata. Consequently, soil water consumption decreases, leading to an increase in soil moisture content (positive effect on soil moisture content, SMC). This increase in SMC offers a degree of mitigation against drought exposure, referred to as water-saving benefits (negative effect on drought induced mortality, DIM). Simultaneously, eCO$_2$ enhances photosynthesis, resulting in greater carbon gains (positive effect on canopy leaf mass and woody NPP, wNPP). This enhanced tree growth also intensifies competition, leading to natural self-thinning (positive effect on

competition induced mortality, CIM). While the reduction in drought exposure through higher SMC offers a reduction of DIM, the availability of 'more carbon available to lose' under eCO$_2$ contributes to increased drought-induced biomass carbon loss by drought mortality (negative effect of DIM on AGB). The combined effects of enhanced carbon gain, reduced drought exposure, and higher biomass density influence positively or negatively net biomass carbon change.




Given the unavailability of Free Air $CO_2$ Enrichment (FACE) experiments and the scarcity of in-situ measurements within the complex Amazon rainforest, employing process-based modelling emerges as a promising approach for investigating how $eCO_2$ influences the accumulation and loss of biomass. Koch et al (2021) demonstrated that models participating in Coupled Model Intercomparison Project 5 (CMIP5) and CMIP6 can reproduce the response of carbon gains

attributed to tree growth in response to environmental drivers but were rather unable to reproduce the carbon losses observed in inventory data. This model shortcoming primarily results from the fact that CMIP5 and CMIP6 models do not include processes related to tree competition and that (most of them) ignore climate induced mortality processes, although a new generation of global models is under development to address these research gaps (Eller et al., 2020; Koven et al., 2020; Yao et al., 2022). Yu et al (2022) underscored this issue and showed that biomass loss due to tree mortality was overestimated in

Dynamic Global Vegetation Models (DGVMs) when compared to historical forest inventory measurements, and that incorporating observation-based constraints into these models would lead to a reduction in carbon sink predictions by the end of the century. Therefore, conducting modelling studies with realistic representation of tree mortality and incorporating observation-based constraints are crucial steps toward achieving a more reliable projection of the evolution of the Amazon intact forests carbon stocks.


As the representation of mortality in most process-based models is based on prescribing a fixed loss fraction of standing biomass stocks (Adams et al., 2013), there is a clear need for a more realistic simulation of tree mortality-induced carbon loss. In Yao et al (2023), an empirical hydraulic failure and a light competition tree mortality module in the ORCHIDEE land surface model were tested over the Amazon rainforest. This model has been calibrated to reproduce tree-

size dependent mortality rates at Caxiuana (a long term throughfall exclusion experiment) and proven effective in reproducing the increasing carbon loss due to tree mortality rate and the resulting basin-scale deceleration in the net carbon sink observed by inventories from Hubau et al. (2020).

In this study, we explore the impact of $eCO_2$ on forest growth, tree mortality, and drought recovery in the Amazon

rainforest. Our analysis leverages the newly-upgraded process-based model ORCHIDEE-CAN-NHA (r7236) with competition and drought induced mortalities, following the methodology outlined in Yao et al (2023). We conducted two factorial simulations, one with rising $CO_2$ levels since 1901, and one without, respectively. The three key specific questions we address here are as follows: (i): does $eCO_2$ lead to a greater increase in tree mortality compared to productivity? (ii) does $eCO_2$ promote carbon loss more during wet years compared to dry years? (iii) does $eCO_2$ alleviate the impact of drought on

net AGB carbon balance and benefit drought recovery? Our hypothesis is that $eCO_2$ leads to a lesser increase in tree mortality compared to carbon gain, and this net benefit for AGB changes is greater during dry years, contributing to accelerated drought recovery. The process-based model ORCHIDEE-CAN-NHA has been well-calibrated. In brief, this model incorporates a mechanistic plant hydraulic architecture simulating water potentials at half-hour intervals within the



soil-root-stem-leaf continuum. It also includes a drought exposure-related tree mortality scheme and accounts for size-
dependent tree mortality rates under exposure conditions. For detailed model description, calibration information and
validation against observed datasets, please refer to Yao et al (2022) and section 2.

## 2 Materials and methods

### 2.1 Model description

ORCHIDEE-CAN-NHA incorporates a plant hydraulic architecture that enables the modelling of water potential
and hydraulic conductance along the vertical profile of plants. This module considers both vertical water flow driven by
water potential gradients and the movement of water into or out of water storage pools, regulated by water capacitance. By
simulating the plant hydrodynamics, we derive a critical indicator of plant water stress: the percentage loss of conductance
(PLC). PLC has been demonstrated to correlate with tree mortality (Choat et al., 2012), like $\psi_{50}$, which represents the water
potential at which 50% of conductance is lost. Building on the simulation of $\psi_{50}$, we have integrated an empirical tree
mortality module that is based on drought exposure, which can help reproduce the size-dependent tree mortality pattern of
higher tree mortality rate in cohorts with larger circumference class. Within this framework, two crucial empirical
parameters have been introduced: the drought exposure threshold and the fraction of tree mortality once this threshold is
reached. These two parameters were calibrated using observed water potentials, sap flux transpiration and stem mortality
rates from long-term throughfall exclusion experiment conducted at the Caxiuana site located in northeastern Amazon (Yao
et al., 2022). The calibrated model has proven accurate for capturing the sensitivity of carbon fluxes to drought and the long-
term trends in net carbon sink dynamics, in comparisons of the simulated sensitivity of biomass loss rates and growth rates to
water deficit against plot observations for the droughts of 2015 and 2010 (Yao et al., 2023). Besides such a drought-induced
tree mortality, ORCHIDEE also parametrizes the light competition-induced self-thinning process (Joetzjer et al., 2022),
which offers competition-induced tree mortality. The self-thinning process in ORCHIDEE is regulated by the quadratic
mean diameter, where smaller trees are killed in priority.

Our analysis only focuses on carbon gains (CG) and carbon losses (CL) from trees with diameter at breast height
exceeding 10 cm to facilitate the comparison with periodic forest inventory results. Observational time series of carbon
gains, losses, and the net carbon balance for Amazonian forests are obtained from Brienen et al (2015). To gain a deeper
insight into how $eCO_2$ impacts carbon loss, we examined both changes in competition-induced (self-thinning, CIM) and
drought-induced tree mortality (DIM) as distinct components. For drought mortality, we compared the drought exposure
under constant and $eCO_2$, for assessing how $eCO_2$ alleviates the risk of tree mortality from hydraulic failure.

Following a TRENDY-type protocol (Seiler et al., 2022), we have implemented two distinct scenarios in our study.
The first scenario maintains a constant $CO_2$ concentration at the 1901 level but varying climate forcing (A1), while the
second scenario permits variations in both $CO_2$ concentration and climate forcing (A2). Here A2 is similar to S2 in
TRENDY protocol despite that we did not consider land cover change.





## 2.2 Climate forcing data

The gridded climate forcing dataset employed is CRUJRA v2.1 (Harris, 2020) used in the TRENDY simulations. CRUJRA v2.1 was created by re-gridding data from the Japanese Reanalysis Data (JRA), a product of the Japanese Meteorological Agency. This re-gridded dataset was adjusted to align with the monthly observation-based Climatic Research Unit (CRU) TS4.04 data (Harris et al., 2020). CRUJRA v2.1 provides 6-hourly meteorological variables spanning from January 1901 to December 2019, at a spatial resolution of 0.5°.

## 2.3 Drought characteristics

Following Papastefanou et al (2022), for the evaluation of drought area and severity, the maximum climatological water deficit (MCWD) was used to compare droughts, as given by Equations (1) and (2) with a fixed value for evapotranspiration (ET) of ~100 mm per month being used (Phillips et al., 2009). When monthly rainfall ($P_m$) is below 100 mm, the forest undergoes water deficit. Water deficit accumulates over the hydrological year from October in the previous year to September in the current one. MCWD is the most negative value of the water deficit among all 12 months.

$$CWD_m = CWD_{m-1} + (P_m-100) \text{ if } P_m < 100, \text{ else } CWD_m=0 \qquad (1)$$

with m being the month from October to September.

$$MCWD = min(CWD_m), m=1, …, 12 \qquad (2)$$

Then the decadal mean of MCWD over the whole period ($\mu_{MCWD}$) was subtracted from MCWD of a year with drought ($MCWD_i$), giving a MCWD anomaly (Eq. 3).

$$MCWD \text{ } anomaly = MCWD_i − \mu_{MCWD} \text{ } i=1980\text{-}2019. \qquad (3)$$

We derived Z-scores of MCWD time series at annual scale following Equation (4) as in Feldpausch *et al* (2016), according to:

$$Z_{MCWD} = \frac{MCWD_i − \mu_{MCWD}}{\sigma_{MCWD}} \text{ } i=1980\text{-}2019 \qquad (4)$$

$\sigma_{MCWD}$ is the inter-annual standard deviation of MCWD.

## 2.4 Drought resistance and resilience

For each drought event, we defined drought resistance as the relative rate of net biomass carbon change during and before drought disturbance, and drought resilience as the ability of net biomass carbon change to recover to the pre-drought state. The calculation of drought resistance and resilience of net biomass carbon change followed Tao et al (2022). We also used the net biomass carbon balance 2 years before and 2 years after the drought event to represent forest pre- and post-drought conditions, respectively (Tao et al., 2022). Resistance and resilience were calculated for each pixel and for all the drought events during the past four decades following Equations (5) and (6) and were reported at the basin scale by taking the median value across drought-affected pixels ($Z_{MCWD}$ below -1).



$$Drought\ resistance = \frac{Y_e - Y_{pre}}{Y_{pre}} \tag{5}$$

$$Drought\ resilience = \frac{Y_{post} - Y_{pre}}{Y_{pre}} \tag{6}$$

$Y_{pre}$ as the pre-drought value of net biomass carbon change (CG minus CL)

$Y_{post}$ as the post-drought value of net biomass carbon change

$Y_e$ as the signal during the drought event of net biomass carbon change

## 3 Results

### 3.1 The mean carbon gains and carbon losses over the Amazon rainforest

Our model simulations over the past four decades, driven by varying $CO_2$ and climate forcing, reveal that carbon gain (CG) slightly surpasses carbon loss (CL) at the basin scale, resulting in a positive net carbon balance ('Net' in Fig. 2a).
When we separate years into wet and dry categories based on the basin-scale median of Z-transformed MCWD ($Z_{MCWD} > 0$ as wet years, $Z_{MCWD} < 0$ as dry years), we find a net carbon sink during wet years and a net carbon source during dry years. The pattern arises because CG is lower and CL is higher during dry years compared to wet years and vice versa. During dry years, competition-induced carbon loss (CIM) is a bit lower compared to wet years and increased drought-induced tree mortality (DIM) leads to higher CL. Under constant $CO_2$ concentration conditions (A1), both CG and CL decrease compared 200 to A2, with the reduction in CG being more pronounced, resulting in a much smaller net carbon sink than under $eCO_2$ conditions (Fig. 2b). Carbon loss from CIM decreases as well during dry years compared to wet years, and its fraction in the total carbon loss becomes lower, as drought-induced tree dieback increases can suppress self-thinning. Comparing the model simulations with and without $eCO_2$, higher $\Delta$CG during dry years compared to wet years suggests the $CO_2$ fertilization effect is more efficient during dry years (Fig. 2c). $\Delta$CL is primarily affected by CIM, even though $\Delta$CIM and $\Delta$DIM have opposing 205 effects on it (Fig. 2c). Our model simulation thus implies that the $CO_2$ fertilization effect plays a dominant role in augmenting forest aboveground productivity (carbon gains) and to a lesser extent biomass loss rates from mortality.

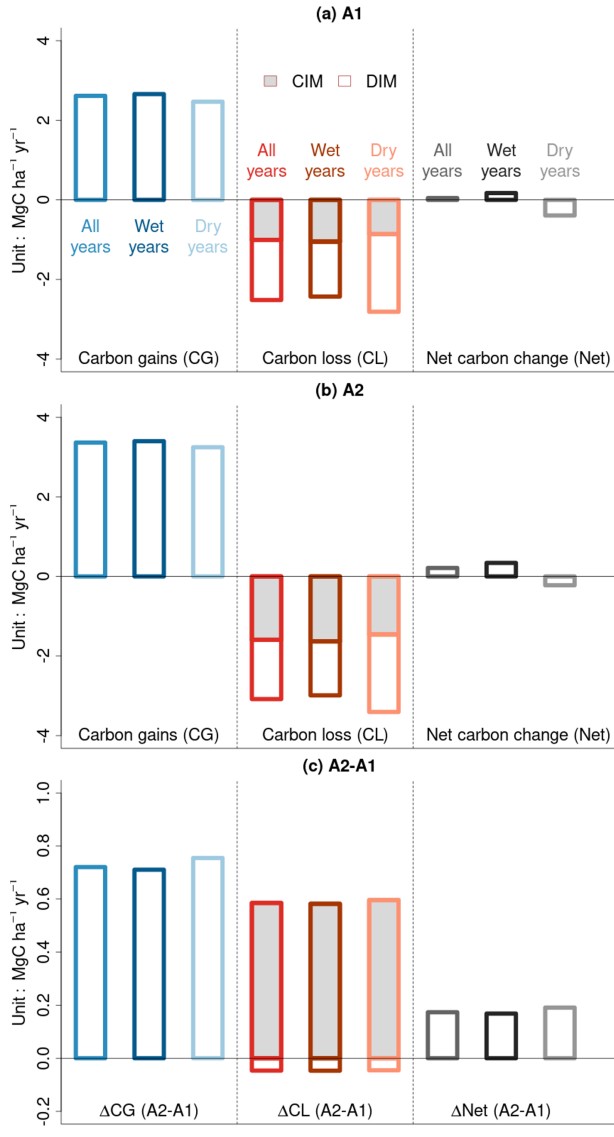

**Figure 2.** Basin-scale average of forest biomass carbon gains (CG), carbon losses (CL) and the net balance of gains minus losses (Net) over the last four decades. (a) $eCO_2$, (b) constant $CO_2$ at pre-industrial level, (c) difference between $eCO_2$ and constant $CO_2$ (Δ). By convention gains are positive and losses are negative in (a) and (b). In panel (c), ΔCL is calculated by the difference between absolute values of CL in panels (a) and (b). Dry years are defined as those in which the median of Z-transformed cumulative water deficit ($Z_{MCWD}$) at the basin scale fails below 0. For biomass carbon loss, CIM represents competition-induced self-thinning processes, and DIM represents drought-induced tree mortality processes.



## 3.2 Effect of eCO₂ on the trends of biomass net carbon sink, carbon gains and carbon losses

Our simulation that accounts for varying climate and $CO_2$ concentration (A2) produces a decline in the net biomass carbon sink since 1980, declining at a rate of 0.006 MgC ha$^{-1}$ yr$^{-2}$ (6 kgC ha$^{-1}$ yr$^{-2}$). This decelerating trend can be predominantly attributed to the increase in carbon loss resulting from tree mortality, which amounts to 0.014 MgC ha$^{-1}$ yr$^{-2}$, surpassing the enhanced carbon gain trend of 0.008 MgC ha$^{-1}$ yr$^{-2}$. The trend of the biomass sink in A2 has the same sign as in forest inventory records, but it is 60% smaller in magnitude (see Fig. 3). When $CO_2$ concentration is held constant, the A1 scenario indicates a larger decline in the net carbon sink. This more negative trend is primarily driven by reduced carbon gains, while carbon loss increases less.

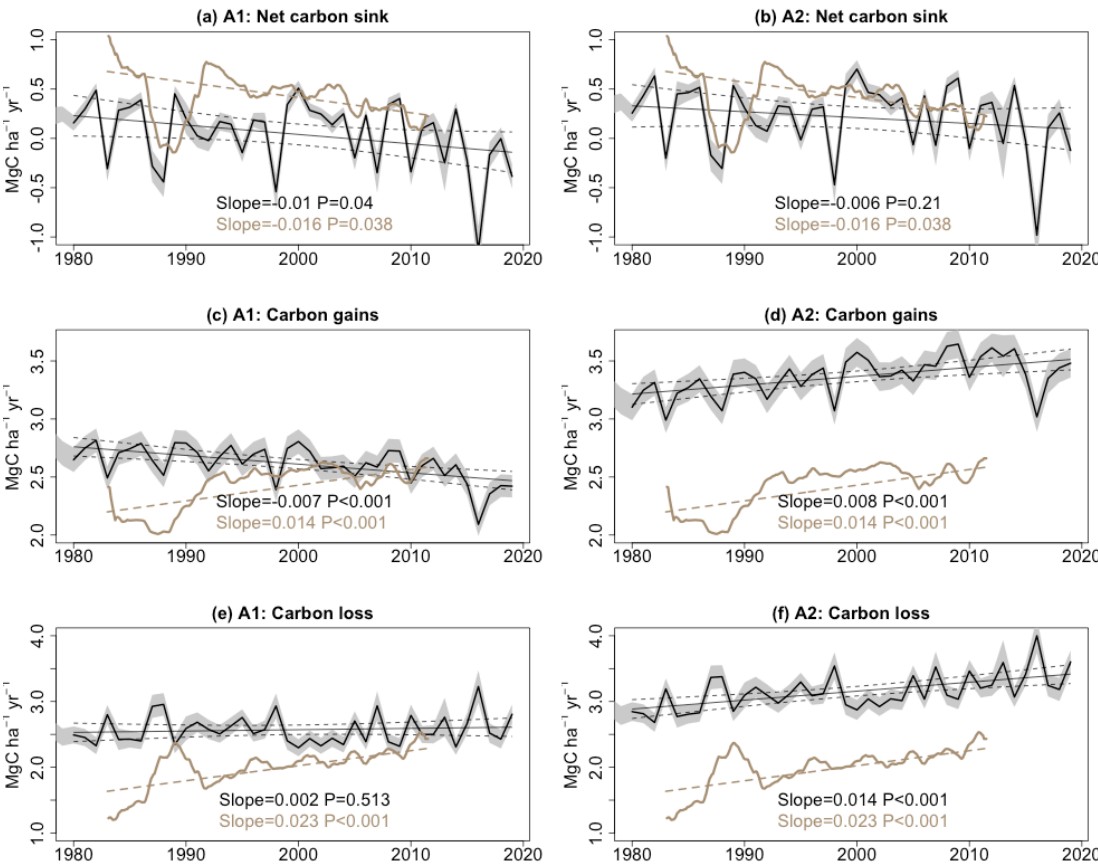

**Figure 3.** Trends in (a, b) net aboveground live biomass carbon sink, (c, d) carbon gains, and (e, f) carbon losses from tree mortality, obtained from ORCHIDEE model simulations (black lines) and forest inventory data (brown lines). Shading represents the 95% confidence interval. The slopes and associated P values are from linear regression models. It should be noted that the number of measurements for each year in inventory varies and 'linear mixed-effects model' was used to



account for the weight associated with different sampling plot areas and their monitoring time length. Therefore, in inventory
pooling results, the trend for net carbon sink is not equal to the difference between trends in carbon gain and carbon loss.

### 3.3 Effect of eCO$_2$ on competition-induced and drought-induced carbon losses

As described in Section 3.1, carbon loss resulting from tree mortality can be categorized into two distinct processes:
competition-induced (CIM) and drought-induced (DIM). Figure 4 illustrates the trends in carbon loss attributed to these two
processes. In the A2 scenario, the simulated CIM displays no significant trend (slope=0.001 MgC ha$^{-1}$ yr$^{-2}$, P=0.738).
However, when CO$_2$ remains constant in A1, this term exhibits a notable decrease (slope=-0.01 MgC ha$^{-1}$ yr$^{-2}$, P<0.001). In
contrast, both A1 and A2 exhibit significant increasing trends in DIM (slope=0.013 MgC ha$^{-1}$ yr$^{-2}$, P<0.05). Consequently,
the lack of a significant overall trend in total carbon loss in A1 can be attributed to the opposing effects of CIM and DIM.
Our model simulations reaffirm that the increasing carbon loss in the A2 scenario is primarily a result of a higher drought-
induced tree mortality. Without the sustained increase in CO$_2$, the net carbon sink would have diminished at a faster pace
(slope=-0.01 MgC ha$^{-1}$ yr$^{-2}$ in A1 vs. slope=-0.006 MgC ha$^{-1}$ yr$^{-2}$ in A2).

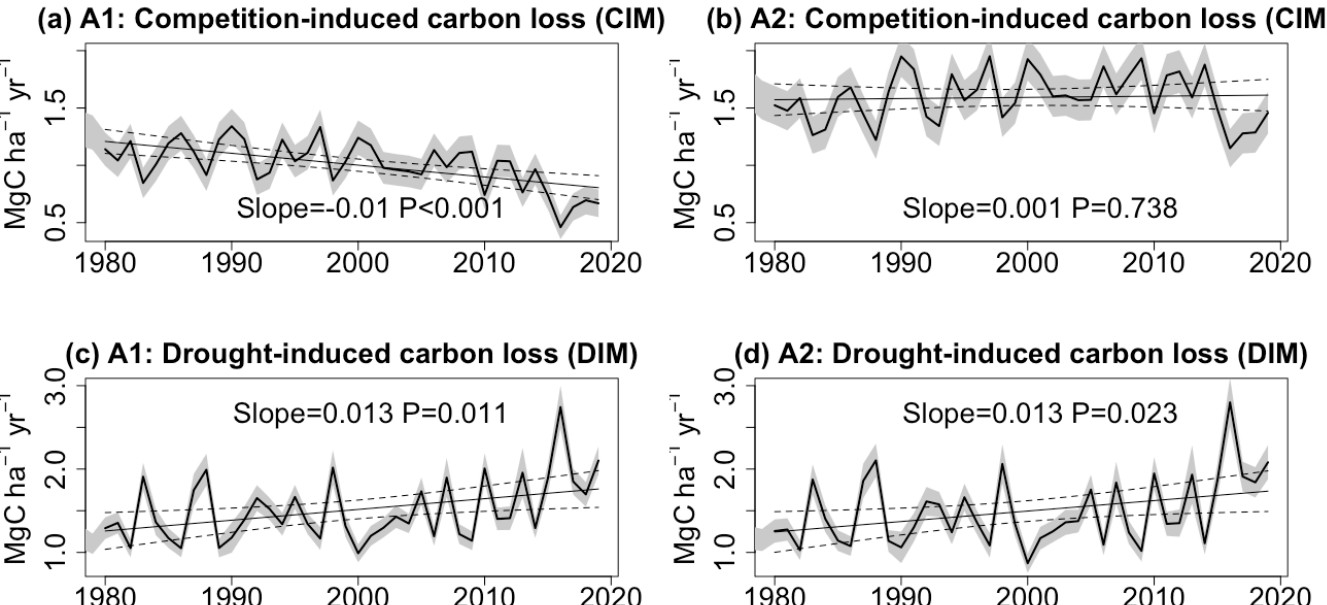

**Figure 4.** Trends in carbon losses due to (a,b) competition-induced (CIM) and (c,d) drought-induced tree mortality (DIM)
with and without eCO$_2$.



### 3.4 Water conditions mediate biomass carbon fluxes responses to eCO$_2$

The impact of eCO$_2$ on biomass carbon sinks is influenced by prevailing water conditions. To explore how hydro-climate conditions regulate the impact of eCO$_2$ on carbon gain and carbon loss, we focused on three recent mega-drought events (2005, 2010, 2015/2016). In line with the methodology applied by Pan et al (2022) to eCO$_2$ vs. control experiments, we used the ratio between enhanced CO$_2$ (A2) and the constant CO$_2$ scenario (A1) to assess the relative response (R) of ecosystems for carbon gain ($R_{CG}$) and carbon loss ($R_{CL}$). During these three drought events, we found that forests in drier climate zones (more negative MCWD) exhibited greater $R_{CG}$ compared to their wetter counterparts (Fig. 5), and this model response prevails across all cohorts, with larger-sized cohorts showing lower $R_{CG}$ and less negative sensitivity of $R_{CG}$ to water deficit of MCWD due to more carbon allocation to smaller cohorts (Fig. S1). Interestingly, $R_{CL}$ does not show monotonic change from small to large cohorts even though the average over the drought epicenter indicates higher $R_{CL}$ in smaller cohorts (Fig. S2). Self-thinning may not always occur due to the suppression by DIM. Therefore, even though higher $R_{CG}$ is found under eCO$_2$ along the water stress gradient, self-thinning does not always change coordinately. Here, $R_{CL}$ is mainly contributed by drought-induced carbon loss ($R_{CL\_DIM}$), where the self-thinning is suppressed. However, $R_{CL\_DIM}$ does not exhibit a significant correlation with MCWD (Fig. 5). It's worth noting that DIM-induced carbon loss is influenced by two key factors: background biomass density and tree mortality rate. The former one is boosted by eCO$_2$, indicating 'more carbon available to lose' (Fig. 5), while the response of the latter is the opposite because eCO$_2$ leads to a reduction in stomatal conductance and transpiration, alleviating water stress, shown by less drought exposure days (Fig. S3) and lower fraction of trees killed due to DIM in the A2 scenario compared to the A1 scenario across most regions within the epicenter of drought events ($Z_{MCWD} < -1$, Fig. 5). This suggests that eCO$_2$ has a positive impact on mitigating the effects of drought on biomass loss driven by DIM.





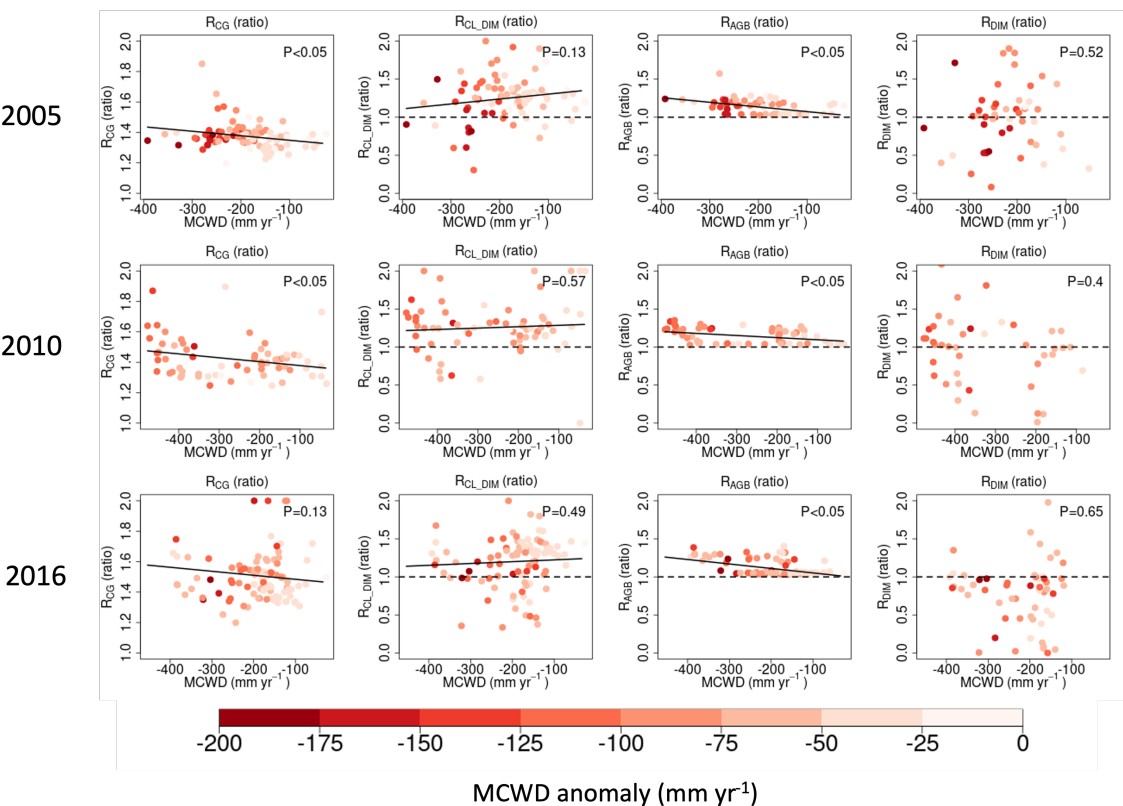

**Figure 5.** The effect of eCO$_2$ on carbon gain (R$_{CG}$), carbon loss due to drought induced tree mortality (R$_{CL\_DIM}$), AGB (R$_{AGB}$), and the proportion of trees affected by DIM (R$_{DIM}$) in relation to water conditions and drought intensity in the years 2005, 2010 and 2016. Water conditions are characterized by MCWD during the drought year on the horizontal axis, where negative values indicate drier climate. Dots are color-coded to reflect the drought intensity characterized by MCWD anomaly, with darker colors indicating more severe water deficits. The dots shown in the panel correspond to pixels located in the epicenter of the drought, featuring Z$_{MCWD}$ values below -1. This threshold is set to ensure an adequate number of pixels for our analysis.

## 3.5 Effect of eCO$_2$ on drought recovery

After a drought, eCO$_2$ supports faster or more complete recovery due to growth enhancement (De Kauwe et al., 2021). eCO$_2$ can also enable deeper root depth, thus a better access to deeper soil moisture (Iversen, 2010). In addition to investigating the impact of eCO$_2$ on carbon loss during droughts, our study delved into whether eCO$_2$ confers benefits to drought recovery considering the likely promoted growth and two different tree mortality regimes of CIM and DIM. To identify past drought events, we calculated the MCWD for each year and each pixel. Droughts were defined as pixels in a year with a Z score of MCWD falling below -1 since 1980. This threshold, while not extremely severe, was chosen to ensure




enough pixels for analysis. Following the methodology outlined by Tao et al (2022), we aggregated the pixel-level results to the basin scale by calculating the median value across all pixels. Considering the findings of Tao et al (2022), which
indicated increased drought resistance across a gradient of drought severity using C-band radar signals, we would see if the process-based model can reproduce this drought response. Figure 6 illustrates the relationship between drought resistance and resilience of net biomass carbon change in relation to drought severity expressed as $Z_{MCWD}$. In both the A1 and A2 scenarios, we observed a well-expected declining trend in resistance as drought severity increased. When drought severity was defined using absolute value of MCWD anomaly, a similar declining trend of resistance for net carbon change was
found (Fig. S4). Notably, the rainforests did not fully recover to their pre-drought conditions following drought events of higher severity (A1: 62±12% vs. A2: 59±16% of the area located in drought epicenter showing resilience below 0). When examining the differences in drought resistance and resilience between the A1 and A2 scenarios, our findings indicate that $eCO_2$ enhances both drought resistance and resilience. For drought sensitive areas where the tree mortality regime shifts from CIM to DIM, their drought resistance and resilience are lower than that of insensitive areas due to higher carbon loss.
This suggests that elevated $CO_2$ levels contribute to improved forest resilience and their ability to withstand and recover from drought events.



**Figure 6.** The effect of eCO$_2$ on drought recovery in aspects of resistance and resilience of net biomass carbon change. For
each drought event, the median resistance, resilience and drought severity of the drought-affected pixels (Z$_{MCWD}$ below -1)
were calculated, shown as dots in each panel. Drought severity was defined by the Z$_{MCWD}$. The size of the dots indicates the
area situated within the drought epicenter. The dots for 2005, 2010, and 2016 were color-coded in red. A trend fitting was
applied to panels a-d, with its equation labelled and 95% confidence bound shaded. Resistance was computed, where lower
values mean a more pronounced reduction in net biomass carbon sink during a drought. Resilience was computed, where
positive values indicate complete recovery of the forest to pre-drought conditions, while negative values signify incomplete
recovery. In panels a-d, a horizontal dashed line indicates a resistance and resilience value of zero. In panels e-f, a horizontal
dashed line was drawn as well to denote no change in resistance or resilience under eCO$_2$ conditions.





## 4 Discussion

The $CO_2$ fertilization effect has gained widespread recognition as a primary driver of vegetation greening observed as an increase of LAI in most regions around the world (Zhu et al., 2016). Our model forced by $eCO_2$ also captures the LAI greening in most areas over the basin, while a constant $CO_2$ setup results in browning (Fig. S5). However, it is important to clarify that this greening phenomenon doesn't necessarily translate into increased biomass accumulation because the carbon allocation shift and processes that control biomass loss can be decoupled from the increasing trend of LAI and productivity

(Fan et al., 2022). The dynamics of the net carbon sink are fundamentally shaped by the balance between productivity and carbon loss. The concept of a growth-mortality trade-off or "high gain high loss" has been observed across spatial gradients (Wright et al., 2010), but it remains more difficult to prove in terms of temporal changes expressed as "faster gains imply higher losses" because one can observe an acceleration of both gains and losses like Hubau et al (2020) but a causal relation between the two can't be empirically proven. A demography model including more detailed stand dynamics can help test

this hypothesis. $eCO_2$ enhancing on tree competition in wet tropical forest has the potential to couple a faster gain to a higher loss, and thus may not universally lead to a boost in the net carbon sink due to an accelerated carbon turnover rate (Walker et al., 2021). Future $CO_2$ fertilization could potentially increase not only recruitment and growth, but also tree mortality (McDowell et al., 2020). In our study, we conducted model experiments to disentangle the effect of $eCO_2$ on forest biomass carbon fluxes, including carbon gain, carbon loss, and net carbon balance. Our model simulations revealed a compelling

insight. When we deactivated the $CO_2$ fertilization effect, the model simulated a LAI browning trend (Fig. S5) and a declining trend in carbon gain compared to forest inventory observations (Fig. 2). This finding aligns with a similar compensatory effect on LAI from rising $CO_2$ in TRENDY models in the tropical forests, as noted in Winkler et al (2021). Our results also underscored that $eCO_2$ doesn't solely drive enhanced biomass carbon gains; it also plays a pivotal role in shaping carbon losses (Figs. 3, 4). Specifically, turning off the $eCO_2$ effect leads to a dampened increase in carbon loss. This

attenuation is primarily attributable to the muted response of natural self-thinning related tree mortality, determined by less growth, that is the deceleration of carbon gain. Meanwhile, the more negative net carbon sink trend derived in scenario A1 is primarily driven by the deceleration of carbon gain. Basin-scale average shows that CL_DIM is lower under $eCO_2$ but CL_ST is highly promoted under $eCO_2$. Thus $\Delta$CL_DIM and $\Delta$CL_ST oppose each other, although $\Delta$CL_ST dominates the magnitude of $\Delta$CL (Fig. 1c). Such net forest biomass carbon loss in the absence of $eCO_2$ was similar to the finding in de

Almeida Castanho et al (2016) using multiple model simulations. They showed a decline in biomass over the last several decades when not considering $eCO_2$ effects, although the tree mortality has not been incorporated yet. In total, our work steps further to separate the contributions of competition-induced and drought-induced tree carbon loss and provides evidence that $eCO_2$ benefits the forest biomass net carbon sink over the Amazon rainforest, although such benefit still may not be sufficient to offset the carbon loss caused by prolonged external climate stressors, like the long-term temperature

induced carbon loss (Sullivan et al., 2020).





The response of biomass carbon flux dynamics to $eCO_2$ is intricately linked to water availability. Taking the years 2005, 2010, and 2016 as examples, the enhancement ratio of carbon gain ($R_{CG}$) exhibits a negative correlation with the water deficit metric of MCWD, indicating that the $eCO_2$ effect is more pronounced in drier regions. Pan et al (2022) demonstrated that extra-tropical woody ecosystems characterized by drier baseline climates tend to exhibit a higher average enhancement in aboveground carbon gain in response to $eCO_2$. Our model simulations suggest that a similar pattern may persist in wet tropical forests within the Amazon as well. The negative interactions found between $eCO_2$ and water deficit suggests that vegetation in drier climates can benefit more from the combination of enhanced photosynthesis, reduced photorespiration, and higher WUE. Alongside potential growth advantages, the tree-ring width data analysis by Zuidema et al (2020) reveals that $eCO_2$-induced partial stomata closure and reduced transpiration may attenuate the cooling effect on leaf surfaces, potentially pushing leaf surface temperatures beyond the optimal range for photosynthesis. Our modelling results, showing enhanced carbon gain over the Amazon rainforest, suggest the temperature may not have exceeded the optimal range in this case given the mean annual land surface temperature under $eCO_2$ increases almost 0.26°C at most during 2015-16 El Nino (Fig. S6). Nonetheless, the potential shift in sensitivity within the carbon flux response of tropical forests to $eCO_2$ depending on the balance between benefits and potential temperature-related challenges, emphasizes the necessity for optimizing and refining the process-based representation of climate-growth relationships. For instance, employing various sensitivity scenarios involving temperature intervals and $CO_2$ concentrations can offer valuable insights for identifying the critical threshold beyond which the benefits of $eCO_2$ cannot outweigh the adverse high temperature effects.

While self-thinning induced carbon loss is heightened in the $eCO_2$ scenario because of increased competition, drought-induced carbon loss might not always be exacerbated under $eCO_2$ although there is more biomass built up over time that is available for carbon loss, revealing the influence of water saving conditions on carbon loss. An important indicator we employed in our model, the cumulative drought exposure, reveals a reduction over most areas when accounting for the water-saving effects driven by $eCO_2$ (Fig. S3). Regarding tree mortality triggered by drought events, our modelling work confirms the alleviated drought exposure, consistent with previous modelling findings that $eCO_2$ will bring about water saving due to increased WUE, thus enhance vegetation productivity and decrease the probability of forest dieback in the eastern Amazon basin threatened by drier and warmer climate scenarios (Huntingford et al., 2013; Lapola et al., 2009; Zhang et al., 2015), although these studies mainly considered the enhanced productivity in sustaining the biomass rather than carbon loss changes. The strength of this mitigation effect depends on the intensity and duration of stress (Lapola et al., 2009) and could postpone the point at which forests shift from being a carbon sink to a carbon source (Feng et al., 2017). The decreased frequency of tree mortality risk, combined with an increase in background biomass stock facilitated by $eCO_2$, however, contribute to uncertainty regarding the fate of carbon loss.

In our modelling study, we found an increase in both WUE and tree growth (Figs. 1, S7). While van der Sleen et al (2014) reported no growth stimulation of tropical trees by $CO_2$ fertilization, they did find an increase in WUE in their tree



ring width analysis, focusing on a fixed tree size class. However, it has been argued by Brienen et al (2017b) that this approach, aimed to remove the effect of tree size, might lead to biased interpretation of growth trends, particularly when there is a clustered age distribution with the coexistence of fast-growing and slow-growing trees. Concerning the less controversial increase in WUE, Brienen et al (2017a) suggested that the observed trends could be related to developmental

effects rather than being solely the result of climate and $CO_2$ effects on WUE. It is crucial to investigate and distinguish these factors, as WUE varies with tree developmental stages, especially in broadleaf forests. The partial stomatal closure, driven by increasing constraints in water transport with tree height and increasing photosynthesis due to greater light availability with tree height, can lead to changes in intrinsic WUE (McDowell et al., 2011). To better isolate the effects of external factors like $eCO_2$ through size-stratified sampling and account for varying tree gas regulation strategies throughout a tree's

lifespan, it is essential to incorporate a stratified simulation of stomatal conductance, as well as corresponding photosynthesis. While our ORCHIDEE model features a stratified LAI pattern, it does not yet include stratified simulation of stomatal conductance. Implementing a more detailed water budget per canopy layer would provide a more comprehensive understanding of tree height-related shifts in WUE.

Regarding drought resistance and resilience, C-band radar data has demonstrated a decrease in resistance to drought over the Amazon rainforest during the past three decades, while forest resilience did not decline significantly (Tao et al., 2022). Our model simulation also detected a well-expected phenomenon of decreased resistance with increasing drought severity. The resistance and resilience of net biomass carbon balance were found to benefit from $eCO_2$, which is broadly consistent with the enhanced resistance to drought due to restricted stomatal conductance and improved WUE in Feng et al

(2017). This suggests that $eCO_2$ can enhance the recovery of ecosystem carbon uptake after short-term drought events.

Several uncertainties warrant in-depth investigations. Whether $eCO_2$ would lead to biomass growth also depends on the carbon allocation strategies subsequently the carbon turnover rate (Friend et al., 2014; Hofhansl et al., 2016), which has been found to be governed by hydraulic constraints, such as the hydraulic adjustment of the ratio between leaf area and

sapwood area (Trugman et al., 2019). Given that both tree productivity and mortality responses during drought are sensitive to hydraulic traits (Anderegg et al., 2016; 2018), incorporating varying hydraulic traits adaptive to the environment will be highly important (Madani et al., 2018). For example, tree mortality risk is intricately linked to plant water use strategies, with isohydric tree species exhibiting a lower xylem embolism risk due to their tendency to close stomata earlier to conserve water. In contrast, anisohydric tree species, characterized by less conservative water use strategies, may derive more

significant benefits from $eCO_2$-induced partial stomatal closure. Additionally, the analysis of water deficit affiliation has indicated that genera affiliated with wetter climate regimes exhibit a higher risk of drought-induced tree mortality (Esquivel-Muelbert et al., 2017). Exploring the interactions between $eCO_2$ and varying hydraulic vulnerabilities would be a potential avenue for further examining the effects of $eCO_2$ on biomass carbon dynamics. Besides the hydraulic failure induced tree mortality, other possible sources including carbon starvation should be included as more carbon gain enhanced by $eCO_2$





would delay the depletion of carbohydrate reserves. Furthermore, it's essential to consider the legacy effects of drought, a dimension that has not been addressed in process-based modelling. Yang et al (2023) used a first-order kinetics model to account for the gradual decomposition of coarse woody debris, yielding a better correspondence between net biomass carbon change and variability in atmospheric $CO_2$ growth rate. The legacy effects from tree mortality should be carefully revisited, given the evidence suggesting that external drivers can lead to increased mortality for at least two years after a climatic event

(Aleixo et al., 2019). Regarding the strength and persistence of $eCO_2$, previous studies have suggested that such fertilization effects could slow down (Penuelas et al., 2017), and the $eCO_2$ effect has declined in recent years, possibly due to nutrient limitation (Winkler et al., 2021). Wieder et al (2015) demonstrated that accounting for nitrogen and nitrogen-phosphorus limitation lowers projected productivity and could even turn terrestrial ecosystems into carbon sources. Fleischer et al (2019) highlighted the important role of phosphorus acquisition and use strategies in regulating forest response to $eCO_2$, reducing

the expected stimulation otherwise by 50% over the Amazon rainforest. The lack of downregulation on fertilization in the model could lead to an overestimation of $eCO_2$ effects. Therefore, estimating the strength and persistence of the $CO_2$ fertilization effect under future climate scenarios remains challenging (Nolte et al., 2023). Additional observations are imperative, and the AmazonFACE project will be a strong observational constraint on our knowledge of the rainforest response to $eCO_2$ (Lapola and Norby, 2014).


In our study, we conducted offline simulations and found that $eCO_2$ leads to an increase in WUE (Fig. S7), which could partially mitigate drought risk through soil-atmosphere feedback mechanisms. However, it's noteworthy that $CO_2$-induced physiological effects reduce ET and subsequently precipitation in land-atmosphere coupled mode. A recent study using coupled climate model simulations has highlighted that the reduction of ET under $eCO_2$ and its impact on

precipitation, contribute to potential water stress (Li et al., 2023; Skinner et al., 2018). Tree dieback indeed leads to reduction in plant transpiration, but also decreases the soil moisture consumption. We found $eCO_2$ leads to an increase of 0-0.26°C in land surface temperature based on the simulation during 2015-16 El Nino (Fig. S6). Therefore, given the contribution of moisture recycling to precipitation over the Amazon rainforest, a comprehensive investigation of the effects of $eCO_2$ on biomass carbon dynamics, like whether $eCO_2$ can mitigate the negative effects of water stress due to changes in

precipitation, should be conducted in a coupled mode including tree mortality module to capture the intricate interactions among these components.

## 5 Conclusions

In summary, this work offers a comprehensive basin-scale quantitative assessment of how $eCO_2$ influences

aboveground biomass carbon gain and carbon loss in a warming and increasingly water-stressed climate. We systematically disentangle the effect of $eCO_2$ in this complex ecosystem. Our findings not only underscore the role of $eCO_2$ in shaping the 'high gain high loss' pattern but also highlight its water saving benefits. Additionally, we identify an enhancement in drought





resistance and resilience attributed to eCO$_2$, as it accelerates drought recovery. Our results offer valuable insights into ecosystem response to eCO$_2$ that cannot be easily obtained through field experiments alone. These results serve as a

compelling impetus for further modelling and observational work aimed at gaining a deeper understanding of the role of eCO$_2$ in predicting the forest biomass carbon budget within the Amazon rainforest.

**Code availability**

The ORCHIDEE-CAN-NHA model (r7236) code used in this study is deposited at

https://forge.ipsl.jussieu.fr/orchidee/browser/branches/publications/ORCHIDEE_CAN_NHA (last access: 17 June 2021) and archived at https://doi.org/10.14768/8C2D06FB-0020-4BC5-A831-C876F5FBBFE9 (Yao, 2021).

**Competing interests**

The authors declare that they have no conflict of interest.


**Author contribution**

PC and YY designed the study. YY ran the simulation, analyzed the outputs and drafted the manuscript. All authors contributed to the final manuscript.

**Acknowledgements**

This work was financially supported by the CLAND Convergence Institute funded by ANR (16-CONV-0003). YY also acknowledges support from Make Our Planet Great Again (MOPGA) Scholarship.

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
