# Peer review of "The impacts of elevated CO2 on forest growth, mortality and recovery in the Amazon rainforest"

_Earth System Dynamics, 2024_

## Author Comment (AC1)

Author's response to the Reviewers' comments

Please refer to the detailed itemized responses below. Our responses are indicated in blue text and the edits are highlighted in red text.

Reviewer #1

[Comment #1] The study uses a new version of the ORCHIDEE model to study elevated CO2 impact on forest growth and mortality in the Amazon in the past decades. The model was previously calibrated at several Amazon sites and was applied at regional scale with and without historical CO2 increase. The simulations with elevated CO2 can better reproduce the temporal trend of C gain and C loss estimated from long-term field plots. Comparison between the simulations with and without CO2 effects show that elevated CO2 increased both growth and mortality while the latter is caused by increased competition and elevated CO2 reduced drought-induced mortality. Further spatial analysis reveals that the CO2 effect is stronger in drier regions. Overall, it is neat to use a model to separate the processes (CIM and DIM) over the Amazon. The manuscript is clear and well written. I feel the mortality response makes sense but I am not sure how much we should trust simulated growth responses to eCO2 as outlined below.

Response #1:

We thank the Reviewer for the time and effort to thoroughly evaluate our study and appreciate the Reviewer's constructive comments. We have added comparisons to existing studies on $eCO_2$ and expanded the discussion on the uncertainties associated with $eCO_2$. We believe we have effectively addressed the concerns raised by the Reviewer.

[Comment #2] I am concerned that models overestimated average carbon gain and carbon loss by ~30% or more (3.0-3.5 Mg/ha/yr vs observed 2-2.5 Mg/ha/yr, Fig.3) in simulation A2 but not in A1. To me, this means simulations with elevated CO2 greatly overestimated baseline growth (and thus mortality), suggesting the CO2 fertilization effect might be overestimated. It would be important to explain this difference in baseline values.

Response #2:

The overestimated baseline growth (and mortality) likely results from nutrient limitations that are not modeled or other model structural errors. In particular, uncertainties in carbon allocation may contribute to differences in baseline values compared to inventory. In the ORCHIDEE model, carbon allocation among biomass components follows the 'pipe model' theory, which determines the relationship between leaf area, sapwood area and fine root area (Sitch et al., 2003). However, the carbon allocation process is relatively unconstrained and requires further observation for benchmarking. Given that nutrient availability influences productivity and carbon allocation adjustments, a nutrient-enabled version of the model would help better elucidate ecosystem responses to $eCO_2$.

The explanation on the possible overestimation of baseline growth rates can be found as follows.

> In addition to the absence of downregulation due to nutrient availability, uncertainties in carbon allocation could also contribute to differences in baseline values compared to inventory data. In the ORCHIDEE model, carbon allocation among biomass components adheres to the 'pipe model' theory, which dictates the relationship between leaf area, sapwood area and fine root area (Sitch et al., 2003). However, the carbon allocation process remains relatively unconstrained and requires further observation data for benchmarking

purposes. Given that nutrient availability influences productivity and adjustments in carbon allocation, a nutrient-enabled version of the model would help elucidate ecosystem responses to $eCO_2$. Therefore, estimating the strength and persistence of the $CO_2$ fertilization effect under future climate scenarios remains challenging (Nolte et al., 2023). Additional observations are imperative, and the AmazonFACE project will be a robust observational constraint on our knowledge of the rainforest's response to $eCO_2$ (Lapola and Norby, 2014).

**[Comment #3]** In addition, the positive trend of carbon gains in observation is mainly due to increase from 1980s to early 1990s. I believe the trend is much weaker after 1990s and in the same Hubau et al. study, there was not growth trend in Africa, suggesting CO2 fertilization effect on growth is quite uncertain. For instance, van der Sleen et al. 2014 reported no growth simulation by CO2 from tropical tree rings. More recently, Jiang et al. 2020 reported eCO2 increased GPP but not woody NPP in an Eucalyptus woodland. Such allocation changes are briefly mentioned in Discussion (line 415 - 425) while I think it should be highlighted as one of the major limitation/uncertainty of the study. For example, how would your conclusion change if the CO2 effect on growth is overestimated by 50% - 100%?

Response #3:

In Hubau et al (2020), the observed positive trend of carbon gains is indeed higher in the earlier period of inventory (before 1993: 0.029 MgC ha$^{-1}$ yr$^{-1}$) compared to the later stage (after 1993: 0.009 MgC ha$^{-1}$ yr$^{-1}$), although the relatively smaller number of monitored plots in the earlier period may also contribute to this difference. We acknowledge that the $eCO_2$ fertilization effect remains subject to large uncertainties. Particularly, the impact of $eCO_2$ on woody NPP is influenced by both nutrient limitation and carbon allocation strategies.

For growth response to $eCO_2$, we summarized existing studies on the $eCO_2$ effects below (Table R1), including process-based model approaches, analytical solutions and ecological optimality theory. In our simulations, the effect of $eCO_2$ on carbon gains (AGB gains before mortality) is estimated to be approximately 5% per decade. The increasing trend in carbon gains derived from inventory data is calculated to be 0.014 MgC ha$^{-1}$ yr$^{-1}$, which equates to an increase of almost 6.2% per decade. This trend reflects contributions from various factors, including the effects of $eCO_2$, climate change, nutrient limitation and other factors. Disturbance recovery is probably not important for the plot data as they are undisturbed forests. Therefore, if negative climate effects are assumed, the 'intrinsic' $eCO_2$ effect should be slightly higher than the 6.2% value derived from inventory data. Hence, our model estimate of 5% per decade, falling within the upper range of the existing trend distribution, is not unreasonable.

We made revisions in the Results to describe the comparison with other existing $eCO_2$ studies. Please see the text as follows.

Our model simulation thus implies that the $CO_2$ fertilization effect plays a dominant role in augmenting forest aboveground productivity (carbon gains) and to a lesser extent biomass loss rates from mortality. Our estimate falls within the upper range of trend distribution, which is consistent with existing studies on the effects of $eCO_2$, including those employing process-based models, analytical solutions and ecological optimality theory (Table S1).

We made revisions in the Discussion to highlight that the $eCO_2$ effects embedding in our model could subject to overestimation given the non-explicit consideration of nutrient limitations and uncertainties associated with biomass carbon allocation.

The lack of downregulation on fertilization in the model could lead to an overestimation of $eCO_2$ effects. In addition to the absence of downregulation due to nutrient availability, uncertainties in carbon allocation could also contribute to differences in baseline values compared to inventory data. In the ORCHIDEE model, carbon allocation among biomass components adheres to the 'pipe model' theory, which dictates the relationship between leaf area, sapwood area and fine root area (Sitch et al., 2003). However, the carbon allocation process remains relatively unconstrained and requires further observation data for benchmarking purposes. Given that nutrient availability influences productivity and adjustments in carbon allocation, a nutrient-enabled version of the model would help elucidate ecosystem responses to $eCO_2$. Therefore, estimating the strength and persistence of the $CO_2$ fertilization effect under future climate scenarios remains challenging (Nolte et al., 2023). Additional observations are imperative, and the AmazonFACE project will be a robust observational constraint on our knowledge of the rainforest's response to $eCO_2$ (Lapola and Norby, 2014).

Table R1 Summary of $eCO_2$ fertilization effects.

| Time period | Term | Magnitude | Method | Reference |
|---|---|---|---|---|
| 1980-2019 | AGB gain (DBH>10cm) | Amazon rainforest: 5% per decade | ORCHIDEE model with climate impacts on growth and mortality, $CO_2$, stand level demography | This study |
| 2001-2016 | GPP | Global 4.1% per decade  EBF: 4.8% per decade | Analytical approach | Chen et al (2022) |
| 2001-2016 | GPP | EBF: 1.61-5.78% per decade | TRENDY models (S1) | Chen et al (2022) |
| 1981-2020 | GPP | Global: 3.4% per decade | Remote sensing + ecological optimality theory | Keenan et al (2023) |
| 1982-2011 | NPP | Tropical: 2.7% per decade | CMIP5 | Smith et al (2016) |
| 1980-2016 | GPP | Tropical: 3.7% per decade | CABLE model | Haverd et al (2020) |

**[Comment #4]** Finally, since AmazonFACE is mentioned, it would be interesting to provide results from some short-term (e.g. 5-10 years, single site) simulation results using similar magnitude of CO2 increase. This can serve as a priori estimate of AmazonFACE results (not necessarily correct).

Response #4

Thanks for your suggestions. We agree with the importance of having a prior estimate for such a FACE experiment. The AmazonFACE experiment is situated in the Amazon rainforest near Manaus, Brazil. We conducted a short-term simulation focusing on the Manaus site, where $CO_2$ will be artificially elevated by 200 ppm above ambient levels. The simulations were conducted for the period from 2010 to 2020, considering two scenarios: one forced by ambient $CO_2$ concentration and the other forced by elevated $CO_2$ concentration (ambient + 200 ppm).

The discussion section has been revised as follows.

> Additional observations are imperative, and the AmazonFACE project will be a robust observational constraint on our knowledge of the rainforest's response to $eCO_2$ (Lapola and Norby, 2014). We have also provided estimates of carbon gain and carbon loss in response to the planned $CO_2$ increase (i.e. 200 ppm above ambient levels) at this forest for the period from 2010 to 2020. Our simulations indicate an enhancement of ~34% in GPP and ~55% in woody NPP (DBH>10cm) throughout the simulation period. These values are higher compared to simulations conducted with nutrient cycle-enabled models as reported by Fleischer et al (2019). Obtaining more experimental data to illustrate the interactions between water and nutrient availability and their impacts on the $CO_2$ fertilization effect would aid in constraining model responses, thus enabling more accurate predictions of the Amazon rainforest's response to future climate change.

References

Chen, C., Riley, W. J., Prentice, I. C., & Keenan, T. F. (2022). CO2 fertilization of terrestrial photosynthesis inferred from site to global scales. *Proceedings of the National Academy of Sciences*, *119*(10), e2115627119.

Fleischer, K., Rammig, A., De Kauwe, M. G., Walker, A. P., Domingues, T. F., Fuchslueger, L., ... & Lapola, D. M. (2019). Amazon forest response to CO2 fertilization dependent on plant phosphorus acquisition. *Nature Geoscience*, *12*(9), 736-741.

Haverd, V., Smith, B., Canadell, J. G., Cuntz, M., Mikaloff-Fletcher, S., Farquhar, G., ... & Trudinger, C. M. (2020). Higher than expected CO2 fertilization inferred from leaf to global observations. *Global Change Biology*, *26*(4), 2390-2402.

Keenan, T. F., Luo, X., Stocker, B. D., De Kauwe, M. G., Medlyn, B. E., Prentice, I. C., ... & Zhou, S. (2023). A constraint on historic growth in global photosynthesis due to rising CO2. *Nature Climate Change*, *13*(12), 1376-1381.

Kolby Smith, W., Reed, S. C., Cleveland, C. C., Ballantyne, A. P., Anderegg, W. R., Wieder, W. R., ... & Running, S. W. (2016). Large divergence of satellite and Earth system model estimates of global terrestrial CO2 fertilization. *Nature Climate Change*, *6*(3), 306-310.

Sitch, S., Smith, B., Prentice, I. C., Arneth, A., Bondeau, A., Cramer, W., ... & Venevsky, S. (2003). Evaluation of ecosystem dynamics, plant geography and terrestrial carbon cycling in the LPJ dynamic global vegetation model. *Global Change Biology*, *9*(2), 161-185.

---

## Author Comment (AC2)

Author's response to the Reviewers' comments

Please refer to the detailed itemized responses below. Our responses are indicated in blue text and the edits are highlighted in red text.

Reviewer #2

[Comment #1] In this study, Yao et al. used a well-established ecosystem model equipped with plant physiology, demography, and hydraulic processes to simulate the carbon sink response to CO2 fertilization in the Amazon rainforest. The results in the figure and texts are well presented, and the experiment simulations are reasonable. While I enjoy reading this work, I found that the paper needs to extract more clear messages especially in the Abstract and Conclusion. For example, what do we learn from this advanced improvement of the model process related to mortality and hydraulic resistance to droughts, and what does this imply for the carbon cycling in Amazonia? The message is not totally clear to me though detailed results have been reported.

Response #1:

There has been less emphasis on understanding carbon loss compared to productivity changes in response to rising $CO_2$, making it crucial to comprehend how carbon loss varies with changing environmental conditions. Our study distinguishes between carbon losses induced by competition and those induced by drought, as these two types of tree mortality respond differently to their respective drivers. The refinement of our model processes related to mortality and hydraulic resistance to drought will contribute to understanding how the carbon balance changes in response to $eCO_2$, including productivity enhancement as well as changes in carbon loss induced by tree mortality from two distinct schemes.

Following the reviewer's suggestions, we have carefully revised the abstract and conclusion part. Compared to the previous version, we highlight the implications of model advancement.

Abstract:

The Amazon rainforest plays a crucial role in global carbon storage, but a minor destabilization of these forests could result in considerable carbon loss. Among the external factors affecting vegetation, elevated $CO_2$ ($eCO_2$) levels have long been anticipated to have positive impacts on vegetation, including direct photosynthesis / productivity enhancement and increasing water use efficiency. However, the overall impact of $eCO_2$ on the net carbon balance, especially concerning tree mortality-induced carbon loss and recovery following extreme drought events, has remained elusive. Here, we use a process-based model that couples physiological $CO_2$ effects with demography and drought mortality / resistance processes. The model was previously calibrated to reproduce observed drought responses of Amazon forest sites. The model results, based on factorial simulations with and without $eCO_2$, reveal that $eCO_2$ enhances forest growth and promotes competition between trees, leading to more natural self-thinning of the forest stands, following a growth-mortality trade-off response although the growth outweighs the tree loss. Additionally, $eCO_2$ provides water-saving benefits, reducing the risk of tree mortality during drought episodes, although extra carbon losses still could occur due to $eCO_2$ induced increase in background biomass density, thus 'more carbon available to lose' when severe droughts happen. Furthermore, we found that $eCO_2$ accelerates the drought recovery and enhances drought resistance and resilience. By delving into the less-explored aspect of tree mortality response to $eCO_2$, the model improvements advance our understanding of how the carbon balance responds to $eCO_2$ particularly concerning competition-induced continuous carbon loss vs. drought-induced pulse carbon loss mechanisms. These findings provide valuable

insights into the intricate ways in which rising $CO_2$ influences forest carbon dynamics and vulnerability, offering critical understanding of the Amazon rainforest's evolution amidst more frequent and intense extreme climate events.

Conclusion:

In summary, this work offers a comprehensive basin-scale quantitative assessment of how $eCO_2$ influences aboveground biomass carbon gain and carbon loss in a warming and increasingly water-stressed climate. We systematically disentangle the effect of $eCO_2$ in this complex ecosystem. Our findings not only underscore the role of $eCO_2$ in shaping the 'high gain high loss' pattern but also highlight its water saving benefits. Additionally, we identify an enhancement in drought resistance and resilience attributed to $eCO_2$, as it accelerates drought recovery. Our improved model, which separates tree mortality schemes into competition-driven and drought-driven mechanisms, offers a more comprehensive understanding of carbon fluxes in response to $eCO_2$, a perspective that cannot be solely attained through field experiments. With the likelihood of more frequent and intense drought events in the near future, these findings serve as a compelling impetus for further modeling and observational efforts aimed at deeper insights into the role of $eCO_2$ in predicting the forest biomass carbon budget and ecosystem vulnerability within the Amazon rainforest.

**[Comment #2]** My other minor comments are mainly about clarification issues. In Lines 145-150, since the carbon gain and loss time series are from Brienen et al. (2015), why do you say in the first paragraph of the results that the model simulates these two? How do you get carbon gain and carbon loss from the model output? What are the output variables?

Response #2:

In our model, we are able to simulate both carbon gain and carbon loss, where carbon gain refers to the woody NPP for trees cohorts with a diameter above 10 cm, following the standards established by inventory protocols. Carbon loss corresponds to the reduction in woody biomass for cohorts with a diameter above 10 cm. We conducted a comparison of the time series of carbon gain and loss between model simulations and inventory observations (for undisturbed plots). To enhance clarity, we have revised the methods section to provide a clearer description of the model outputs as follows.

As ORCHIDEE is a cohort-based model, we obtain woody carbon gain, woody carbon loss and biomass carbon pools for 20 cohorts, associated with increasing circumference / diameter classes from small trees to large trees. Carbon gain in our model refers to the woody NPP, specifically for cohorts with a diameter above 10 cm, aligning with inventory protocols. Carbon loss represents the amount of live biomass (with diameter >10 cm) that is transferred to the woody litter pool due to tree mortality, from continuous competition induced mortality (killing small trees) and drought induced pulse mortality events (killing large trees). Then we aggregate the grid-level carbon gain and carbon loss to the basin-level, following the approach used by Brienen et al (2015).

**[Comment #3]** The definitions of drought resistance and resilience are not entirely clear to me. The equations are clear, as in Equations (5) and (6). But what do these metrics imply for drought resistance and resilience? More explanations are needed.

Response #3:

We give more explanation on the meaning of these two metrics. Section 2.4 has been revised as follows.

> For each drought event, drought *resistance* is defined as the change in the net biomass carbon sink during the drought disturbance relative to the pre-drought state. A positive value indicates that drought conditions lead to an increase in the net carbon sink relative to non-stressed conditions, while negative values indicate a decrease in the net biomass carbon sink. A more negative value indicates higher vulnerability. Drought *resilience* refers to the ability of the net carbon sink to recover to the pre-drought state. It is computed as the difference in the net carbon sink between the post-drought period and the pre-drought state relative to the pre-drought period. Positive values indicate full recovery, where the net carbon sink after drought stress surpasses the pre-drought state, while negative values indicate incomplete recovery. A more negative ratio represents a more limited capacity for recovery. The calculation of drought resistance and resilience of net biomass carbon change followed the definitions proposed by Tao et al (2022). We also used the net biomass carbon balance 2 years before, and 2 years after a drought event to represent forest pre- and post-drought conditions, respectively (Tao et al., 2022).

**[Comment #4]** Overall, I think this work is very novel and represents our newest process understanding of the Amazonian carbon sink from CO2 forcing from the perspective of models. But the messages need to be clearer.

Response #4:

We have enhanced the clarity of our results in response to the comments. We believe we have effectively addressed their concerns.